# A systematic approach to inserting split inteins for Boolean logic gate engineering and basal activity reduction

Trevor Y. H. Ho [1,2,3], Alexander Shao[1,4], Zeyu Lu [1], Harri Savilahti[5], Filippo Menolascina [6], Lei Wang[7], Neil Dalchau [4] & Baojun Wang [1,2,3,8 ✉]

Split inteins are powerful tools for seamless ligation of synthetic split proteins. Yet, their use remains limited because the already intricate split site identification problem is often complicated by the requirement of extein junction sequences. To address this, we augment a mini-Mu transposon-based screening approach and devise the intein-assisted bisection mapping (IBM) method. IBM robustly reveals clusters of split sites on five proteins, converting them into AND or NAND logic gates. We further show that the use of inteins expands functional sequence space for splitting a protein. We also demonstrate the utility of our approach over rational inference of split sites from secondary structure alignment of homologous proteins, and that basal activities of highly active proteins can be mitigated by splitting them. Our work offers a generalizable and systematic route towards creating split protein-intein fusions for synthetic biology.

[1] Centre for Synthetic and Systems Biology, School of Biological Sciences, University of Edinburgh, Edinburgh, UK. [2] Hangzhou Innovation Centre, Zhejiang University, Hangzhou, China. [3] College of Chemical and Biological Engineering, Zhejiang University, Hangzhou, China. [4] Microsoft Research, Cambridge, UK. [5] Department of Biology, University of Turku, Turku, Finland. [6] Institute for Bioengineering, School of Engineering, University of Edinburgh, Edinburgh, UK. [7] School of Engineering, Westlake University, Hangzhou, China. [8] ZJU-UoE Joint Research Centre for Engineering Biology, Zhejiang University, Haining, China. ✉email: baojun.wang@ed.ac.uk

nteins are internal protein elements that are expressed as part of a larger precursor protein[1–3]. Upon proper folding, an intein excises itself from the precursor protein and ligates the flanking external proteins (exteins) with a peptide bond. This process is known as protein splicing and it produces a product as if the intein was absent from the original gene sequence. Given this unique property, inteins are a popular choice for near-seamless protein ligations, and they become popular with synthetic biologists and biochemists who seek to engineer and chemically modify proteins of their choice.

Inteins can be further classified into contiguous or split inteins. For the latter, the intein is expressed as two separate peptides that can spontaneously self-assemble and perform protein splicing. Split inteins thus enable reconstitution of separate coding sequences with minimal scarring, reducing the chance of having additionally inserted domains that might compromise the original protein function upon structural reconstitution. For protein chemists, the introduction of split inteins enabled new protein modification techniques including protein semi-synthesis[4,5], labeling[6,7], and circularization[8,9]. For synthetic biologists, split inteins are ideal tools for implementing digital logic. A protein can be divided into two and be fused with a split intein, such that the bipartite fragments remain individually inactive[10], and protein function is not restored until protein splicing occurs. This effectively generates an AND or NAND logic to integrate two biological signals. This strategy has been employed in generating in vivo DNA sensors[11], protein-based logic gates for bio-computation[12,13] and enforcing dual conditions in directed evolution[14]. It can be further used to reduce protein sizes for viral delivery[15] and to free up selection pressures in plasmid maintenances[16–18]. Many contiguous and split inteins were discovered and studied, and our group recently characterized a library of orthogonal split inteins extensively[3]. Yet, despite the availability of inteins and their good performance, the use of inteins in synthetic biology applications still face challenges, hindering their widespread adoption by the community. One major issue is the identification of intein insertion or split sites. Whilst this shares some similarities with the search of a general split site, the presence of an intein introduces an extra layer of complexity—inteins require specific extein junction sequences for efficient splicing[19,20]. The composition of amino acid residues around a chosen split site needs to be carefully considered. Alternatively, one can insert characterized extein junctions at a putative split site, or modify host protein residues around a putative split site to satisfy extein junction requirements. Doing so can risk perturbations to the protein structure and function. By either approach, making educated guesses and then testing split sites on the scale of trial-and-error remains the popular route for researchers, and hence the design space is often sparsely and inefficiently sampled[16,21,22].

Multiple solutions were proposed and reported in the literature to tackle this problem. For instance, the SPLICEFINDER[23,24] streamlined the molecular cloning processes to rapidly integrate inteins at sites chosen by researchers, accelerating the build-test cycles. Computation algorithms were also developed to predict split sites from solved or modeled protein structures. One method searches for flexible regions on protein structures and regions that lack functional conservation, and was demonstrated on inserting the gp41-1 intein on genes encoding antibiotic resistances[17,18]. Another method abbreviated SPELL[25] takes protein structures, calculates split energies and identifies surface-exposed loops that contains low conservation in sequences to predict split sites. Yet, SPELL was designed to split proteins with a pair of chemically inducible dimerization (CID) domains, and so might not be fully compatible with intein insertions.

While these computational methods provide better rationality in testing split sites, they rely on protein 3D structures. Thus, they could suffer from reduced accuracies if structures were built by modeling. This is suboptimal for synthetic biologists, who often deploy novel and exotic proteins that typically receive insufficient attention to warrant structural elucidations. In addition, testing only a few split sites risks missing the optimal sites, and this could jeopardize the overall performance of larger synthetic systems if they comprise split intein-inserted proteins as key components.

Here, to facilitate split site identification for synthetic biology applications, we customize and improve previously described mini-Mu transposon-based approaches[26–30]. We develop a bisection mapping method that involves the fusion of split inteins to bisected host proteins. The technique is applied to five proteins and reveals novel split sites for achieving the AND and NAND logic. We highlight the advantage of using an intein compared to interacting domains in splitting proteins, employ our method to evaluate a single case of split site prediction from protein secondary structural homology, and describe suppressing uninduced activities by splitting highly active proteins. Finally, we also make an attempt to engineer switchable inteins, and demonstrate in principle that a small of degree of post-translational inducibility is introduced into an intein for drug-dependent control of protein function.

## Results

**Designing the IBM workflow for split site screening.** In pursuit of a systematic protocol to search for split sites for inteins, we drew heavily from library approaches, which are well-suited for non-rational protein engineering. We utilized the mini-Mu transposon, bisection mapping[26] (BM) and domain-insertion profiling[27,28] (DIP), and incorporated features from the latter into the former. In brief, the method (Fig. 1a and Supplementary Fig. 1) starts with an in vitro transposition reaction that randomly inserts a BbsI and SapI-flanked transposon into a staging vector, which hosts a slightly trimmed, BsaI-flanked coding DNA sequence (CDS) of interest (Supplementary Fig. 2). This is followed by size selection of the insertion library such that only CDS fragments with insertions will be isolated and ligated into a vector for protein expression. A Golden Gate reaction was then used to irreversibly substitute the transposon with a DNA fragment. The fragment carries a selection marker, a split intein, and transcription and translation initiation elements for carboxyl-lobe (C-lobe) expression. In-frame insertions in the right orientation will thus split a CDS into two with the amino-lobes (N-lobes) and C-lobes of the split intein as fusion partners, under separate control of two inducible promoters. The final library is then screened for individual clones that display functional activities only when the chemical inducers for both promoters are present. The clones are then sequenced at the fusion joints to reveal the split sites.

For the split intein, we first selected the *Ssp* DnaB[M86] intein[31] (thereafter referred as the M86 intein) since it only requires the −1 and +1 extein residues for splicing. These extein residues are incorporated into the substitution insert. Successful splicing of the M86 intein would leave behind a highly predictable four-residue peptide linker at the split site of the original protein (Supplementary Fig. 3). Our method emphasizes the use of an intein in bisection mapping—hence the name intein-assisted bisection mapping (IBM).

**mCherry-M86 intein fusion for demonstrating the IBM workflow.** To preempt potential difficulties in troubleshooting if the IBM workflow returned no functional split sites, we first carried out a proof of principle test utilizing mCherry as the target protein to be split. mCherry has been employed as a reporter for Bimolecular Fluorescent Complementation (BiFC) and two split

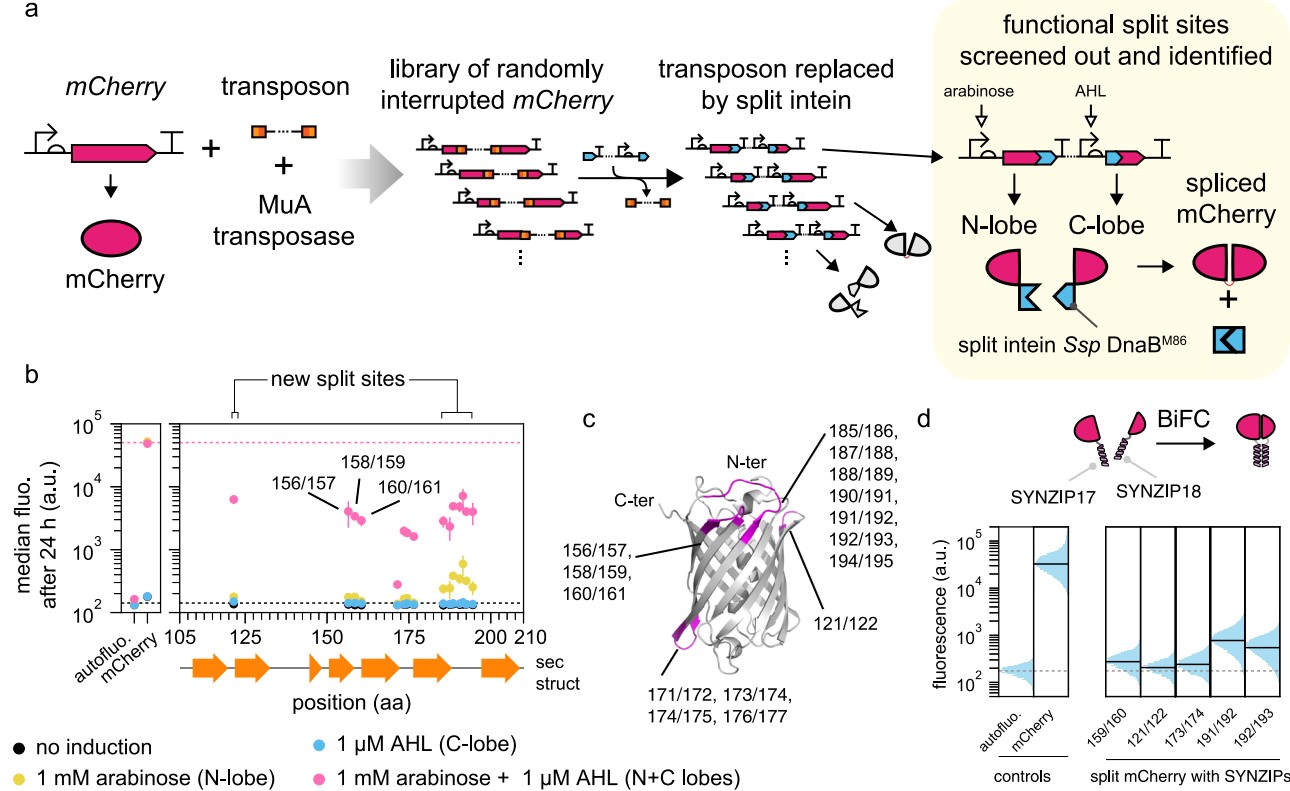

**Fig. 1 Proof-of-concept of Intein-assisted Bisection Mapping (IBM) on mCherry to recover known split sites for BiFC and discover new ones.**
**a** Workflow. A transposon was randomly inserted into the mCherry CDS, and was then substituted with a DNA fragment containing the split intein *Ssp* DnaB[M86]. This generated a library of mCherry-split intein fusions that can be screened for fluorescence only when both the N- and C-lobes were expressed. AHL acyl homoserine lactone. **b** Two new loops for split sites were identified on mCherry and two existing ones were recovered. Each vertical group of spots represents an identified split site, aligned to the mCherry secondary structure below. The majority of sites are between the β-sheets of the barrel. *y* locations and error bars are mean and SD of median fluorescence from experiments performed on three different days ($n \geq 3$, see Supplementary Data 2, sheet "sample_sizes" for exact values of *n*). Horizontal dashed lines bound the range of fluorescence that split mCherry could yield. See Supplementary Fig. 5 for site distributions and activities at 5 h. **c** Split sites mapped to a reconstructed mCherry 3D structure (PDB: 2H5Q). Each split site has the −1 and the +1 amino acid residues colored. **d** Representative split sites from each loop on mCherry permitted biomolecular fluorescence complementation (BiFC). Single-cell fluorescence values were pooled from three biological replicates. Solid black horizontal lines denote population median, except for autofluorescence which was denoted by dotted gray lines.

sites, 159/160 and 174/175, were known to create bipartite lobes that would regain functionality if brought into proximity[32]. We created a split mCherry construct that simulated the known split site 159/160 being sampled by IBM. This construct only gave increased fluorescence when the inputs for N-lobes (inducible by arabinose) and C-lobes (inducible by AHL) were present (Supplementary Fig. 4). Thus, provided enough library coverage, a successful execution of the IBM workflow should generate the simulated control as a member within the final library, and the control should be recoverable afterwards.

Given the assurance, we proceeded to run the IBM workflow on the mCherry protein (Fig. 1a). The resulting library was induced with both arabinose and AHL. Cells with fluorescence above autofluorescence were sorted by fluorescence-activated cell sorting (FACS), plated and isolated as single colonies. Individual strains were then assayed for responses in the absence or presence of the two inducers, and those that showed AND logic behavior were subsequently sequenced to identify the split sites. Pooling the results yielded an intein-bisection map (Fig. 1b and Supplementary Fig. 5). A total of 15 split sites were identified. All split sites clustered into four seams, which were mostly located on loops between the β sheets of the barrel. The second and the third seams should cover sites 159/160 and 174/175, though curiously, site 159/160 was not sampled by this IBM attempt. This

site was present in the library we screened (Supplementary Table 1) and was left out fortuitously, likely due to under sampling. Protein splicing took place at all split sites as evidenced by a Western blot experiment on whole-cell lysates, and we observed that all C-lobes were consumed (Supplementary Fig. 6). This proved that our IBM workflow can locate intein insertion sites that support efficient splicing.

To our knowledge, the seams 121/122 and 185/186–194/195 were never described to contain functional split sites before. Of equal interest was the fact that four split sites, while close to the loops, were found between amino acid residues that constituted the β sheets (Fig. 1c), and this could suggest tolerance of either structural disruption or inserted linkers, both of which are unlikely to be ever tested if split sites were designed rationally. Together, these two observations showcased that IBM has the potential to discover novel and unexpected split sites.

We also noticed that sites within seam 185/186-194/195 yielded a low level of fluorescence when the cells were grown under prolonged induction (24 h) of the N-lobe alone. Other bipartite mCherry split at other seams did not show such behavior. This might be explained by a leaky P$_{lux2}$ promoter, and the fact that mCherry split at the last seam would produce relatively shorter C-lobes that were easier to transcribe and translate.

Since the literature reported split sites on mCherry were developed for bimolecular fluorescent complementation[32], we asked whether the new split sites identified could serve the same purpose. We arbitrarily selected one or two representative split sites from each seam. We also included three additional sites that were found within the β-sheets. For each site, we built split mCherry constructs where the split M86 intein was removed or replaced by a pair of synthetic and heterodimerizing coil-coiled domains, SYNZIP17 and SYNZIP18[33] (Fig. 1d and Supplementary Fig. 7). For all constructs where split sites were on flexible loops, increase in fluorescence could be observed when both lobes were expressed with SYNZIPs, demonstrating that for mCherry, tolerances of the IBM-identified split sites to protein fusion were not unique to the intein. Sites within the β-sheets, except 176/177, did not yield an increase in fluorescence, likely due to structural disruptions to the β-barrel. To test whether the N-lobes and C-lobes could complement without external help from SYNZIPs, we removed SYNZIP17 from the N-lobes. Results showed no increase in fluorescence and Western blots proved that it was not due to the lack of protein expressions. Of note, two of the tested split sites 191/192 and 192/193 produced overall stronger BiFC activities while their C-lobes were more limited in abundance. This further showed that the IBM pipeline can identify the globally optimal sites to maximize performance of a split protein for a defined application.

**IBM with the gp41-1 intein identified a new split site on β-lactamase.** The IBM workflow is modular in design and switching to another intein should be as simple as employing a new substitution DNA fragment during the step of transposon replacement in Fig. 1a. To demonstrate this, we employed the gp41-1 intein[34] with the −2, −1 and +1, +2 minimal extein residues (GY/SS)[35] and passed the TEM-1 β-lactamase (abbreviated as BLA in figures) through the IBM pipeline. We chose this protein for the same reason as mCherry, as it has a solved crystal structure and a well-established split site at 194/196[36] or 195/196[37]. Furthermore, it has an additional split site 104/105 that was computationally predicted and verified[17].

In the screening process, we enriched ampicillin-resistant split constructs through selection and outgrowth. Subsequent sequencing on candidate clones revealed six split sites on two split seams (Supplementary Fig. 8). The first seam, consisting sites 192/193 and 196/197 corresponded to the established site of 195/196. Whereas the second seam (260/261, 261/262, 264/265, and 267/268) represented a previously unreported split location. We did not recover site 104/105, but this could be due to the clones being outcompeted during the enrichment process. Regardless, our results indicated that IBM works with two different inteins.

**Applying IBM to engineer AND and NAND logic gates.** Having established the IBM workflow, we then sought to demonstrate its universality in engineering protein-based logic gates[38–40]. We focused on transcription factors because their responses could be directly converted to an assayable fluorescent output (Fig. 2a). We chose the repressor TetR and its homolog SrpR from the same protein family[41], and an activator, the extra cytoplasmic sigma factor 20 (ECF20)[42]. Each protein was fed into the IBM workflow using the M86 intein and a corresponding intein-bisection map was generated. Three split seams and 32 split sites were found for TetR; 4 seams and 13 sites for SrpR; 3 seams and 17 sites for ECF20 (Fig. 2b–d and Supplementary Figs. 10–12). Most of the split sites for TetR clustered around loop regions between helices from the TetR crystal structure (PDB: 4AC0, Supplementary Fig. 13). It is noteworthy that the same was observed for SrpR and ECF20 even though their shown secondary structures were only

predictions that we generated from JPred4[43]. The performance of the logic gates in the aspect of on and off states strongly depended on the split protein, the split site locations as well as the time elapsed since induction. Across most split sites found in TetR and ECF20, the split proteins would show qualitative NAND and AND gate behavior with good repression and activation strengths at both 5 and 24 h. For SrpR, most split sites yielded NAND gate behavior at 5 h post-induction with observable levels of repression, but at 24 h, expression of C-lobes alone sufficed to evoke a strong repression, rendering the circuit more like a single input responsive gate (Supplementary Fig. 11). This was caused by the accumulation of the N-lobes from leaky $P_{araBAD}$ expressions over time and the NAND behavior could be restored by eliminating the leakiness (Supplementary Fig. 14), proving that neither the N-lobe or the C-lobe alone was capable of repression. Our results thus demonstrated that the IBM workflow is generally applicable, works on proteins with no solved 3D structures, and exhausts most possible sites. This allows researchers to choose the optimal logic gates for their bespoke applications.

**IBM indirectly defined functional boundaries on the ECF20 activator protein.** While screening the colonies for AND gates in ECF20, we observed that some yielded strong activation activities from the expression of N-lobes alone, which emulated the responses of an intact protein and addition of C-lobes did not further improve activities. We thus surmised that they could be truncations at the C-termini and sequenced some of them. Indeed, those split sites were clustered at 178–185 (Fig. 2d) and approximately corresponded to the end of the last helix on ECF20, suggesting that the residues beyond position 178 could be trimmed without a loss of function. In contrast, the first helix was crucial since the first AND gate split site was found immediately after the helix. These two observations suggested that IBM is not only useful for synthetic biology, but has potential applications in general biology, in determining the minimal functional size of a protein.

**Use of an intein expands the range of split sites discoverable in TetR.** Previously reported approaches in bisection mapping for engineering logic gates utilized protein–protein interacting domains like the SYNZIPs as fusion partners[29]. We hypothesized that inteins make a better choice because they would be excised from the splice product, whereas additional domains could exert steric hindrance, especially when the host protein function requires multimerization or interactions with other proteins. To test our hypothesis, a representative site from each split seam identified on the split TetR was selected and the split M86 intein was replaced by SYNZIPs in a similar manner as above (Fig. 3a). Of the three tested sites, site 166/167 showed a stronger level of repression (~4-fold) compared to the other two (~2-fold), despite having the least possible amount of reconstituted protein (Supplementary Fig. 15). The differences in repression strengths between split sites, when SYNZIPs were used, demonstrated their differential tolerances towards additional domains. Whereas in IBM, all three sites showed good levels of repressions that were strong enough to be identified from a single screen. Hence, the use of an intein could enable split sites that might be inaccessible by protein–protein interacting domains, and expands the range of split sites that can be identified for logic gate engineering.

**IBM revealed limitations in inferring split sites from secondary structure alignment.** While IBM discovers split sites exhaustively, it entails more preparation and work than simple guesses and trial-and-errors. Ideally, if the intein-bisection maps on a representative protein of a known family can inform split site

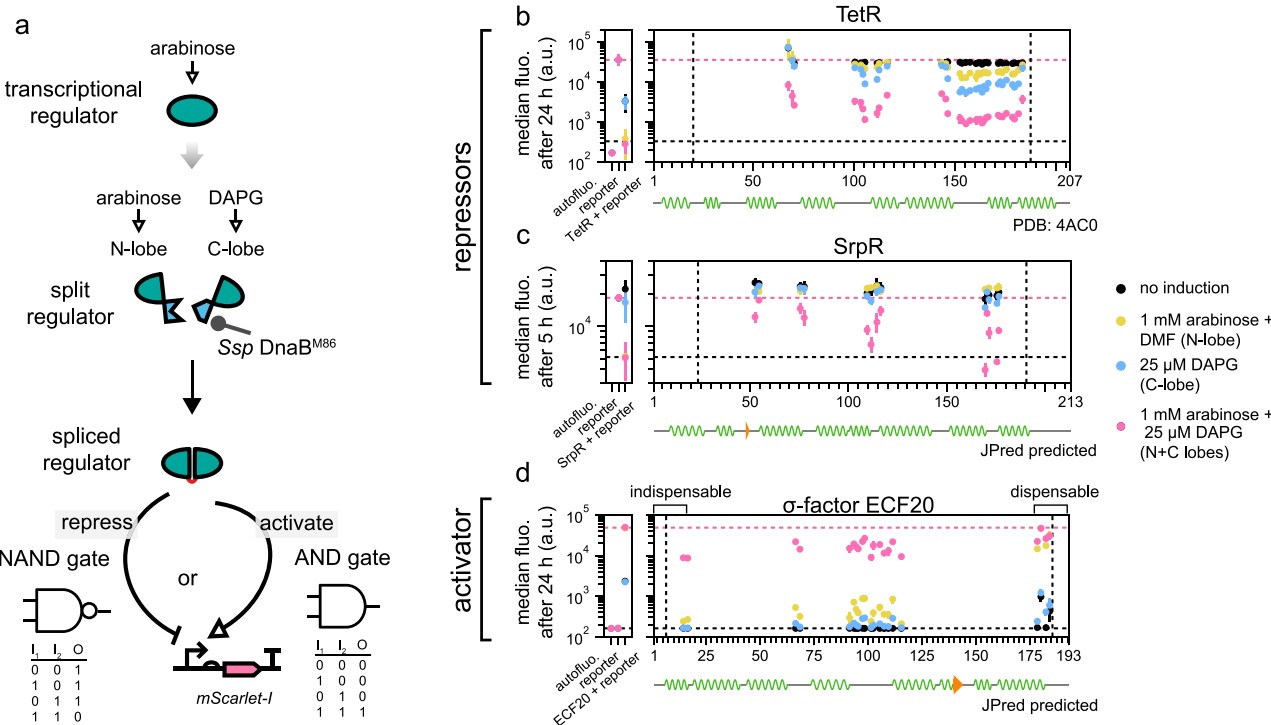

**Fig. 2 IBM as a universal method to exhaust split sites for AND and NAND logic gate engineering. a** Any transcription factor with a function that can be wired to an assay-friendly output could be subjected to IBM for logic gate engineering. **b–d** Intein-bisection maps for TetR (**b**, 3 seams identified), SrpR (**c**, 4 seams), and ECF20 (**d**, 3 seams). Split clones of TetR and SrpR (or ECF) achieved major off (or on) activities when both the N-lobes and the C-lobes were present. *y* locations and error bars are mean and SD of median fluorescence from experiments performed on three different days (*n* ≥ 3, see Supplementary Data 2, sheet "sample_sizes" for exact values of *n*). Vertical dashed lines bound the permitted transposition window and horizontal dashed lines bound the ranges of activities that could be attained by the split proteins. DMF dimethylformamide, DAPG 2,4-Diacetylphloroglucinol. **d** By-products of IBM revealed a truncatable region of ECF20 which when removed did not adversely affect activation. **b–d** See Supplementary Fig. 10–12 for site distributions and activities at 5 h (TetR and ECF20) and 24 h (SrpR). See Supplementary Fig. 9 for explanations of controls (leftmost subplots).

selections on other proteins of the same family, then it might obviate the need to perform IBM on every new protein. Since we had split site data for both TetR and SrpR, both belonging to the TetR family orthologs, we asked how reliable it was to infer split sites assuming only one of the two proteins were bisection mapped. To this end, we took a conventional approach and aligned their sequences, their predicted secondary structures[44], and their split sites for the M86 intein (Fig. 3b and Supplementary Fig. 16). The alignment indicated that only two split seams are shared between TetR and SrpR. Typically, loops that are common to both structures would be assumed to contain putative split sites[45], but on the alignment, sites on seam 2 of TetR and seam 3 of SrpR were mapped to a consensus helical region, and they would be overlooked if split sites were cross-inferred. Likewise, on SrpR split sites were found on the loop that demarcates the DNA binding domain from the dimerization domain, but those sites would be unexpected if TetR served as the reference model. Our results therefore suggest the secondary structure alignment approach might sometimes work, but to ensure guaranteed discovery of insertion sites for split inteins, the IBM approach should be undertaken and the effort would be well-justified.

**Mitigation of undesirable basal activities in highly active proteins through IBM.** Serendipitously, we discovered that splitting highly active proteins could suppress their background activities. This has important implications for split intein applications and protein function control. The observation was best illustrated by the IBM generated split β-lactamases

(Fig. 3c). When the full CDS of β-lactamase was placed under the control of P_araBAD, the hosting bacteria could grow in ampicillin regardless of arabinose addition, proving that leaky expression of β-lactamase was sufficient to confer resistance. In contrast, split β-lactamases did not lead to cell growth if inducers were absent. This tightening of protein expression control could also be concluded from further analyses of single-cell fluorescence data from Fig. 2. Intact TetR and SrpR had stronger repression than their bipartite counterparts at 5 h. At 24 h, however, in the absence of induction the fluorescence of unrepressed cells was much lower (Fig. 3c and Supplementary Fig. 17). Whereas many bipartite repressors at 24 h had higher unrepressed fluorescence, better separation of populations between on and off states, and hence high fold changes. This phenomenon was even more pronounced on the ECF20 activator—at 5 h, basal activities already gave a strong off-state fluorescence and four split constructs started to benefit from fold change improvement, and at 24 h the average fold change of the worst performing split ECF20 construct was around 54 compared to 22 of the intact ECF20 (Supplementary Fig. 17). These data from β-lactamase, repressors, and activators implied that intact proteins had accumulated over time due to leaky expression from a single promoter, but when they were split by IBM and placed under independent promoters, the conferred AND logic led to a lower probability of assembling a functional protein, thereby reducing the overall undesirable basal activities at the off states. Our result pointed to IBM as a potential solution for tightening control over protein functions. This can be valuable in regulating activities of cytotoxic peptides or enzymes whose activities impose strong cellular burden.

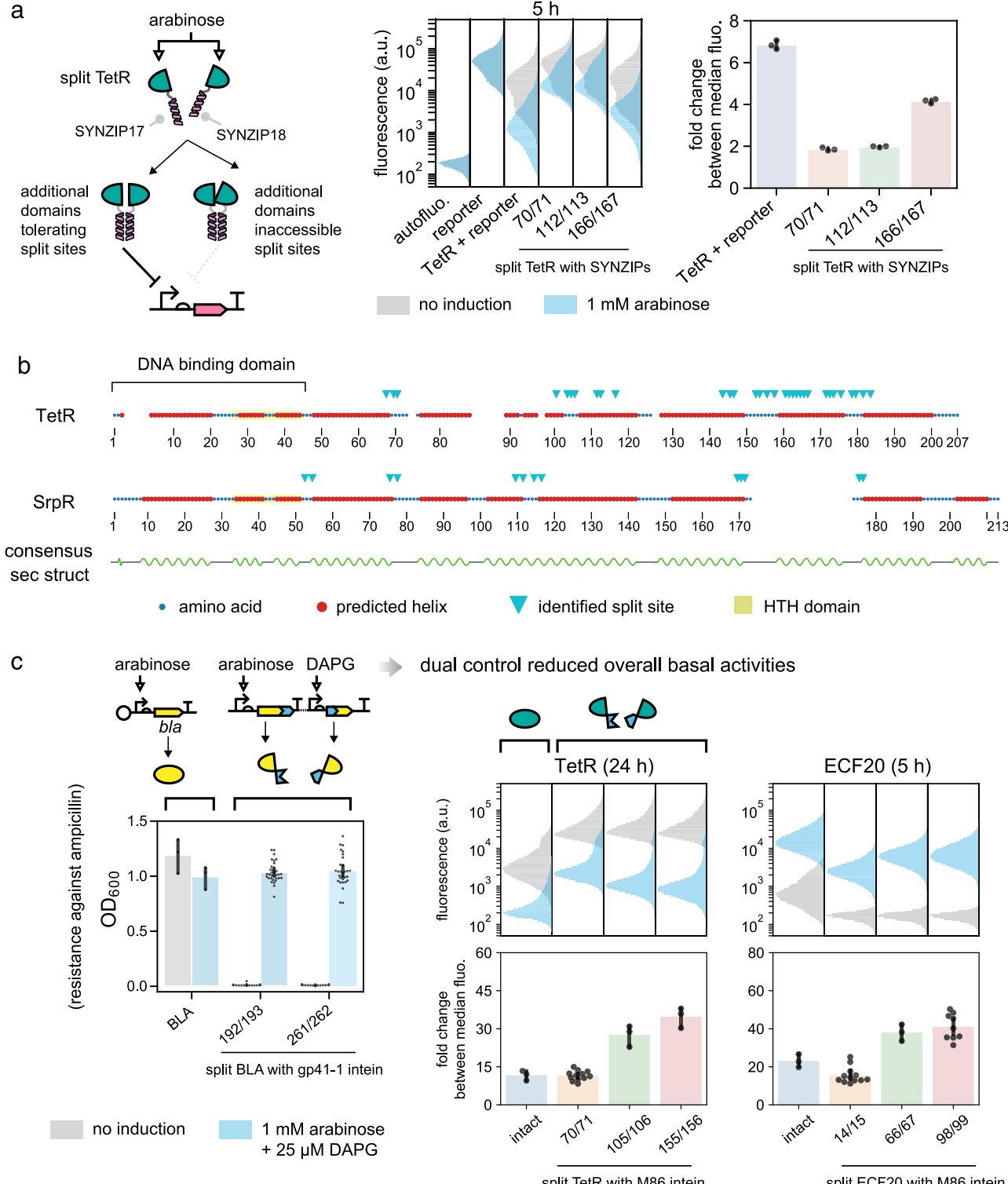

**A proof-of-concept attempt to engineer switchable inteins by transposon-mediated domain-insertions**. In theory, our IBM workflow should allow one to use a conditional intein[46–48] in place of a split intein, and then screen for insertion or split sites that would enable post-translational control of protein function. We thus synthesized and tested three reported chemogenic conditional inteins[49–51] from the literature, but they did not work well under our specific context in *E. coli* (Supplementary Fig. 18). We asked instead whether we could create inducible inteins de novo, by transposing drug-controlled domains into the spontaneous splicing M86 intein (Fig. 4a). We tested two of these domains: the ligand binding domain of human estrogen receptor (ER-LBD)[28], and the camelid anti-caffeine VHH (acVHH) antibodies[52,53].

Insertion of the ER-LBD domain was known to obstruct proper protein folding until the binding of 4-hydroxytamoxifen (4-HT), which then elicits a structural change and relieves the effect[28]. We thus aimed to use this domain to control protein splicing. To do so, we first positioned the contiguous M86 intein between mCherry(1–192) and mCherry(193–236), given that this

**Fig. 3 Benefits of IBM and reduction of basal activities by IBM. a** Substitution of the split M86 intein inserted in split TetR by SYNZIP. Representative sites were chosen within the three identified split seams. Results showed different split sites had differential tolerances towards additional protein domains, but all split sites functioned well when the intein was used (Fig. 2b). Single-cell fluorescence values were pooled from three biological replicates (n = 3). **b** IBM-identified split sites of homologous TetR and SrpR do not necessarily map to loops between consensus helical domains, highlighting the limitation of guessing split sites for a new protein by secondary structure alignment. HTH helix-turn-helix. **c** Splitting highly active proteins can reduce their basal activities. Left panel: leaky expressions of β-lactamase (BLA) led to ampicillin resistance in the absence of induction, which could be improved by splitting BLA. Middle and right panels: fluorescence distributions and fold changes between fully on and fully off states of intact and split TetR and ECF20 were shown. Representative sites were chosen from each identified seam. In most cases the split clones had lower basal activities and therefore larger fold changes between on and off states. Single-cell fluorescence values were pooled from experiments performed on three different days (n ≥ 3, see Supplementary Data 2, sheet "sample_sizes" for exact values of n). DAPG 2,4-Diacetylphloroglucinol. **b**, **c** Data reused from Fig. 2 and Supplementary Fig. 8 but further analyzed. **a**, **c** In fold change calculations, bar heights and error bars represent mean and SD.

mCherry split site performed well in the fluorescence complementation experiment (Fig. 1d). Through our transposition workflow, the ER-LBD was then randomly inserted, and possible insertion sites were limited to the CDS of the contiguous M86 intein only (Fig. 4b and Supplementary Fig. 19). We induced protein expression by the addition of arabinose, performed a series of positive and negative cell sorting with or without 4-HT, then characterized and sequenced candidate strains. Seven insertion sites were identified, and small but distinguishable differential responses between uninduced and induced states could be observed at 24 h. We selected two constructs and subjected them to a gradient of 4-HT induction, then inspected their fluorescence and whole-cell lysates (Fig. 4c). As 4-HT concentration increased, both showed a dose-dependent upshift of fluorescence, and at site 85/86, a gradual increase of spliced product formation could be observed on the Western blot. However, most precursors did not undergo splicing and a low level of splicing happened even when no 4-HT was added. All seven ER-LBD inserted M86 inteins were inserted into site 101/102 of ECF20 but failed to control the activity of the activator (Supplementary Fig. 20).

In a separate attempt, we aimed to engineer chemically inducible dimerization into the M86 intein. We chose to insert the acVHH domains since, in the presence of caffeine, they could homodimerize and reconstitute a split T7 RNA polymerase[54] (Supplementary Fig. 21), and caffeine is inexpensive. We transposed two acVHH domains, each with a 10-residue linker, into the contiguous M86 intein. The intein was already inserted between ECF20(1–101) and ECF20(102–193). The resulting constructs were bipartite proteins, with each part driven by one arabinose inducible promoter. After cell sorting and sequencing, we obtained 12 split sites close to the M86 N-terminus. They had increased fluorescence when cells were grown under 100 μM caffeine for 24 h and being compared to the lack of caffeine induction (Fig. 4d and Supplementary Fig. 22). All positions showed either strong basal and strong maximum activities or weak activities for both states. We characterized two sites, 17/18 and 39/40, for dose-dependent activation under different arabinose concentrations (Fig. 4e and Supplementary Fig. 23). Activation was an increasing function of caffeine and the responses resembled a Hill curve. Reducing arabinose concentration reduced leaky activation at the expense of the overall activities. This could be due to decreased spontaneous assembly rates when intracellular protein concentrations were driven down. We then cloned these two inteins, and an additional intein bisected at 22/23, back into mCherry split at 192/193 to test if they were transferrable, but there were no traces of splicing (Supplementary Fig. 24). We speculated the splicing efficiencies under caffeine induction were too low and only worked with ECF20 given its extreme potency in activation.

Our efforts to create new switchable inteins through domain transpositions have yet to produce practical tools for protein

function control. Nevertheless, we achieved a proof-of-concept demonstration and produced prototypes where imperfect inducibility could be engineered into inteins systematically.

**Discussion**

We have established IBM as a useful tool for split protein-intein engineering. Thus far we have only employed the *Ssp* DnaB[M86] and the gp41-1 intein. Repeating our IBM experiments using different split inteins may reveal even more split sites at new positions, since extein junctions with different amino acid compositions and lengths would be incorporated. This may alter the rigidity of the resulting linkers and hence functionalize other unprobed split sites. If shorter linkers are desired, a promiscuous intein[55] should be helpful, since a designated +2 extein residue can be omitted and supplied by the host protein.

Our IBM workflow addresses issue of split site identification by an empirical approach, and therefore does not face the same constraints encounter by computational methods by Palanisamy et al.[17] and the SPELL algorithm[25]. For split site predictions, sometimes solved 3D structures may be unavailable and de novo structure prediction might not sufficiently reflect the multimerization required in some proteins like TetR. Conversely, IBM requires the protein function to be manifested as an easily screenable output and would be tremendous difficulty or outright impossible if the function of interest is, for example, chemotaxis. It also requires a chassis capable of creating a complex library through highly efficient transformation, and so for the time being it only applies to bacteria or yeasts. These limitations are absent in the computational methods for split site predictions, however. Furthermore, data from IBM may be fed back into and refine those algorithms. We therefore advocate IBM/acVHH-assisted BM not as a competitor, but rather, a complement to the computational approaches.

In this work, we created two inducible variants of the M86 intein. At the current stage, these two inteins have limited dynamic ranges and strong spontaneous splicing activities, and thus are specific to their protein contexts where they were screened. Therefore, our work is a proof of concept, and is only the first step towards a systematic approach in converting spontaneously splicing inteins into conditional ones. So far only the M86 intein was tested as the engineering precursor. Other inteins standardized by our group lately[3] might lead to conditional inteins with better on-off characteristics. There are also other synthetic ligand binding domains, for example, uniRapR[56] and iFKBP/FRB[57] that are potential replacements for ER-LBD and acVHH. Given so, the performance of any inducible intein engineered through this method may depend on complex interactions between the intein of choice, the drug-responsive domain, the insertion/split positions and the host protein itself. Hence, future efforts in engineering more inducible inteins would likely require a combinatorial exploration of all four factors to identify the optimal structure for drug-induced protein splicing, and our

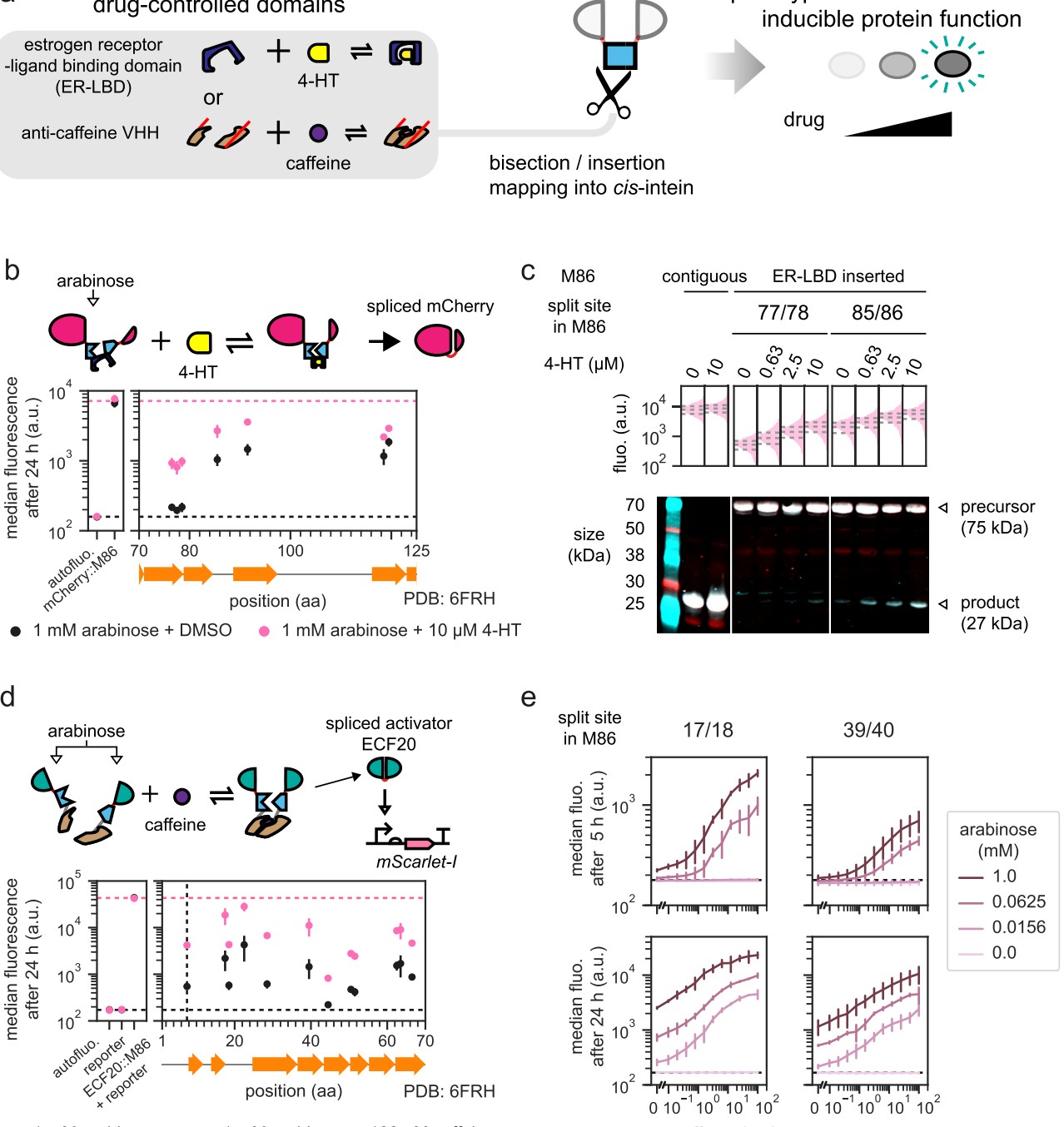

**Fig. 4 Transposon as a tool to engineer inducibility into an intein for protein function control. a** The M86 intein in its *cis*-form can be inserted in an identified split site within the protein of interest, and then bisected or inserted with drug-controlled domains, which leads to inducible splicing and inducible function of the host protein. **b** Seven insertion sites identified via domain-insertion mapping of the estrogen receptor-ligand binding domain into the M86 intein within mCherry interrupted at 192/193, where the addition of 4-hydroxytamoxifen (4-HT) led to increased fluorescence. **c** Selected clones from **b** showed gradual increase in fluorescence and spliced product formation as the concentrations of 4-HT increased, despite the fact that most precursors were unspliced. Single-cell fluorescence values were obtained from one biological sample. Interquartile ranges are denoted by horizontal dashed lines. mCherry N-lobes were labeled in red and C-lobes were labeled in turquoise, and their superpositions give a white color. The result is representative of two independent experiments with similar results. **d** Eleven split sites for anti-caffeine VHH identified on the M86 intein that interrupted the activator ECF20 at 101/102. In those clones, the addition of caffeine led to increased activation activities. **e** Selected clones from **d** had increased activation activities as caffeine concentration increased, and leakiness due to spontaneous assembly could be mitigated by lowering the total split protein concentrations. *y* locations and error bars are mean and SD of median fluorescence of three biological replicates each assayed on a different day (*n* = 3). **b**, **d** Each vertical group of spots represents an identified insertion/split site, aligned to the M86 intein secondary structure. See Supplementary Figs. 19, and 22 for site distributions and activities at 5 h. *y* locations and error bars are mean and SD of median fluorescence from experiments performed on 3 different days (*n* ≥ 3, see Supplementary Data 2, sheet "sample_sizes" for exact values of *n*). DMSO dimethyl sulfoxide. **b**, **d**, **e** Horizontal dashed lines bound the maximum (**b** and **d** only) and minimum fluorescence that could be achieved by the split or inserted constructs.

transposition method should be useful since the components are modular—the same library of an intein being bisected can be cloned into different host proteins, and the transposon can also be swapped out by different substituting domains.

The limited performance of the two inducible inteins from this study could also be improved via directed evolution[21,31,49,58,59], which can mutate these inteins such they splice with better efficiencies and with lower basal activities, and allow them to perform sufficiently well when inserted into other host proteins. We foresee a powerful combination between domain-insertion/acVHH-assisted bisection mapping and directed evolution, where the former identifies the optimal sites for differential activities. The sites can then be exploited by the latter which can use the differential responses to prime the evolution process. This should create more inducible inteins which would be valuable tools when additional domains on a protein cannot be tolerated (Supplementary Fig. 25).

Our IBM workflow employs random transposition to diversify DNA insertion. This method has known issues including sequence bias[60–64] and inexhaustive space search[27,28,64,65]. Despite so, in our final libraries prior to screening, sequenced by Next Generation Sequencing (NGS), we obtained at least 87% coverage on possible amino acid split/insertion positions (Supplementary Fig. 26 and Supplementary Table 1), which sufficiently explored the sequence space. If unbiased and full position coverage is desired, the step of random transposition should be replaced by the recently developed Saturated Programmable Insertion Engineering (SPINE)[64], which hardcodes all insertion possibilities into oligonucleotide pools. Incorporation of SPINE into the IBM workflow should return bisection/insertion maps with higher confidences.

In summary, we presented a robust method to screen and identify protein split sites for the insertion of a split intein. We have recently characterized a library of orthogonal split inteins[3]. With the help of IBM, a large number of orthogonal AND or NAND gates can be created in a streamlined fashion, expediting the development of biocomputing units. Moreover, we demonstrated in principle that transposing drug-controlled domains could create prototype switchable inteins, and that they, to some extent, controlled host protein functions post-translationally. Together, they constitute an empirical and systematic approach towards split protein and protein function control engineering, and should benefit general biologists who seek to use inteins on split proteins.

## Methods

**Strains, media, and inducers**. *Escherichia coli* TOP10 (Invitrogen) was used for routine cloning. For all bisection/insertion mapping, the first insertion libraries were always transformed into the electrocompetent *E. coli* NEB 10-beta (C3020K, NEB). For the rest of the workflow the strain was switched back to TOP10. The only exception was the IBM experiment on mCherry and its outcome simulation experiment, where NEB 10-beta was used in all steps. All strains were grown in the Miller's Lysogeny Broth (LB, 10 g L$^{-1}$ tryptone, 5 g L$^{-1}$ yeast extract, and 10 g L$^{-1}$ sodium chloride) in liquid medium or agar supplemented with the appropriate antibiotics (unless noted otherwise) at the final concentrations of: kanamycin (K4000, Sigma-Aldrich), 50 μg mL$^{-1}$; chloramphenicol (C0378, Sigma-Aldrich), 25 μg mL$^{-1}$; ampicillin (A9518, Sigma-Aldrich), 100 μg mL$^{-1}$; tetracycline (T8032, Sigma-Aldrich), 10 μg mL$^{-1}$; spectinomycin (ab141968, Abcam), 50 μg mL$^{-1}$.

For preparation of stock inducers, powder of L-(+)-Arabinose (A3256, Sigma-Aldrich, 1M), N-(3-Oxohexanoyl)-L-homoserine lactone (AHL, K3007, Sigma-Aldrich, 25 mM), or caffeine (A10431.22, VWR, 10 mM) was dissolved in water; 2,4-Diacetylphloroglucinol (DAPG, 16345, Cambridge BioScience, 25 mM), in dimethylformamide (D4551, Sigma-Aldrich); Rapamycin (S1039-SEL, Stratech, 10 mM), (Z)−4-hydroxytamoxifen (4-HT, H7904, Sigma-Aldrich, 10 mM), 3,3′,5-triiodo-L-thyronine (T$_3$, HY-A0070A, Cambridge BioScience, 10 mM) in dimethyl sulfoxide (D8418, Sigma-Aldrich), with the stock concentrations denoted in brackets. For inductions involving inducers dissolved in organic solvents, the volumes of inducer were less than or equal to 1% of the final volume.

**Molecular cloning**. Synthetic DNA constructs were built using Gibson Assembly[66], Golden Gate Assembly[67] and conventional subcloning using restriction digestion and ligation, with the method chosen depending on their individual needs. Whenever necessary, synonymous mutations were introduced to remove internal BsaI, BbsI, or SapI restriction sites. Standard molecular biology protocols were observed. The ZymoPURE II Plasmid Midiprep Kit (D4200, Zymo) was used for DNA library extractions. For the purification of DNA, the Monarch Nucleic Acid Purification Kits (T1020 and T1030, NEB) were used. All restriction enzymes and ligases were bought from NEB. MuA protein was purified in collaboration with Domus Biotechnologies (Turku, Finland)[68]. List of constructs and list of oligonucleotides used in this study are detailed in Supplementary Data 1.

**Cell growth for fluorescence assays and OD measurements**. Cells were routinely cultured in 96-well plates (655096, Greiner Bio-One) sealed with breathable membranes (Z380059, Sigma-Aldrich), and incubated at 37 °C in plate shakers (AS-03020-00, Allsheng) with 1000 rpm orbital shaking motion. An assay of synthetic constructs began with an inoculation of a single colony from an agar plate into a well with 200 μL of medium, which was then grown for 16–18 h. The next day, 2 μL of the overnight culture was diluted 1:100 into 198 μL of fresh medium with or without inducers and grown for 5 h. The membrane was then removed and 2 μL of the culture was sampled. A new seal was applied, and the plate was returned to the shaker to further grow until the total time of incubation was 24 h. Afterwards, 0.5 μL (24 h) of the culture were sampled. Changes to growth time were noted in individual figures where appropriate. Exception to the above applies to the split mCherry splicing experiment, the mCherry BiFC experiment and the split TetR-SYNZIP experiment, where overnight cultures were diluted 1:100 into 1 mL of fresh medium in 96-deepwell plates (E2896-2110, Starlab). For the assays of strains identified from bisection/insertion mapping experiments, the overnight culture was inoculated from the saved glycerol stocks (described below). Assays measuring resistance against ampicillin were performed in a similar manner, the only difference was that ampicillin was added at the same time as inducers, and growth was only measured after 24 h.

**Optical density measurements by plate reader**. End-point optical densities at 600 nm (OD$_{600}$) were measured with a FLUOstar Omega plate reader (BMG Labtech). The software Omega Control v5.11 (BMG Labtech) was used for data acquisition and Omega MARS v3.32 (BMG Labtech) was used for data export. The optical densities of blank wells from the same plate were subtracted from all other wells.

**Fluorescence measurements by flow cytometry**. Prior to analysis, sampled cultures were diluted into 1× phosphate-buffered saline (K813-500ML, VWR) with 2 mg mL$^{-1}$ kanamycin to a total volume of 200 μL. Diluted cells sampled at 5 h were incubated at 4 °C for a minimum of one hour to promote fluorophore maturation, whereas those at 24 h were directly assayed. Cells were passed into the Attune NxT Flow Cytometer (Thermo Fisher) equipped with the Autosampler for analysis. For each well, 100 μL of diluted cells were run at 500 μL min$^{-1}$ and at least 10$^5$ events were recorded. Red fluorescence was acquired on the YL2-H channel (excitation 561 nm, emission 615/25 nm). Exported FCS files were processed using an in-house Python script dependent on the FlowCytometryTools package v0.5.0. All samples were gated on FCS-H and SSC-H for events between 10$^3$–10$^5$ arbitrary units, followed by gating on YL2-A and YL2-H between 1 to 10$^6$ arbitrary units (Supplementary Fig. 28a). For visualization, the FlowCal package v1.3.0[69] was used.

**Transposition and bisection/insertion library preparations**. A detailed protocol for carrying out IBM is available at protocols.io[70]. The mini-Mu transposon used in this study was modified from the one used by Segall-Shaprio et al.[26] with BbsI and SapI sites incorporated into the R1 recognition sites. Prior to transposition the transposon was released from its host vector by restriction digestion using BglII followed by purification from agarose gels. The coding DNA sequence of interest is trimmed at the N-termini and C-termini before being subcloned into a staging vector. In vitro transposition reactions were set up following an established protocol[71] with slight customizations: 150 ng of the staging plasmid and 150 ng transposon were mixed with 660 ng of MuA. For each insertion library 5–6 reactions of 25 μL each were prepared and incubated at 30 °C in a thermocycler for 6 h, followed by heat inactivation at 80 °C for 10 min. All reactions were pooled, purified, and then eluted in 10 μL of nuclease-free water. The resulting DNA was then electroporated into a total of 200 μL of NEB 10-beta cells in four separate cuvettes and recovered following manufacturer's protocol. Then, 10 μL of the recovered cells (~2 mL) were removed, serially diluted into 0.85% sodium chloride (w/v) and spread onto LB agar with kanamycin and chloramphenicol for colony counting. The library coverage was defined as the total number of obtainable transformants / (size of staging plasmid in bp × 2) and were at least 20-fold for all experiments. Libraries that did not meet the coverage criterion were discarded and transposition reactions were repeated. For libraries with sufficient coverages, the rest of the recovered cells were spread onto LB agar. Bacterial lawns were then washed down by 0.85% sodium chloride and a small aliquot was saved as a glycerol stock. The rest were pelleted, and the DNA was extracted by midiprep.

Ten microgram of the midiprepped DNA from the initial insertion library was digested by BsaI and then resolved on agarose gels until bands were well-separated. The bands corresponding to the trimmed coding DNA sequences with insertions were then excised, purified, and ligated to the linearized expression vector in 1:2 molar ratio for insert:vector. The overnight ligated product was then purified and electroporated into 100 μL of in-house-prepared electrocompetent cells, which were recovered in 2 mL SOC for 1 h, concentrated and then spread onto LB agar and grown overnight. Library coverage estimation and DNA extraction of the library were performed similar to that in transposition, except that the size of the insertable positions equals to the size of trimmed coding DNA sequence in bp. At this stage library coverages were typically >500-fold.

To replace the inserted transposon with split inteins or drug-controlled domains, 60 ng of the midiprepped DNA from the open reading frame (ORF) insertion library was mixed with substitution inserts (released from the cloning plasmids) in a 1:5 molar ratio for plasmid:insert, and added to a Golden Gate reaction mixture[67] with 20 units of BbsI, 10 units of SapI and 400 units of T4 DNA ligase. The reaction was then run in a thermocycler with the following program: (37 °C for 3 min, 16 °C for 4 min) × 25 cycles, 37 °C for 30 min, and 65 °C for 20 min. Usually 5–6 reactions were run, pooled, purified and electroporated into 100 μL of in-house-prepared electrocompetent cells. Electrocompetent cells carried a reporter plasmid wherever required. Cells were recovered and the library coverage estimation was performed in the same manner as the preparation of the ORF insertion library. At this stage library coverages were typically >100-fold.

**Library screening**. For IBM on mCherry, recovered cells from the final library were first induced with both arabinose and AHL, and sorted by fluorescence activated cell sorting (see below). Retrieved cells were then spread onto LB agar for colony picking. For IBM on β-lactamase, recovered cells from the final library were first induced with arabinose and DAPG overnight, the culture was then diluted 1:100 in fresh medium containing arabinose, DAPG and ampicillin, and was grown for another overnight. The resulting culture was serially diluted onto solid medium with inducers and ampicillin for isolating single colonies. For IBM on TetR, SrpR, and ECF20, recovered cells from the final library were serially diluted such that single colonies could be observed when they were spread onto LB agar with arabinose and DAPG. For TetR and SrpR, functional reconstitution of the bipartite protein represses expression of mScarlet-I and therefore yielded visibly white or pale pink colonies. The opposite was true for ECF20. These colonies were picked directly. For the M86 intein inserted with ER-LBD or the M86 intein bisected by acVHH, recovered cells from the final library were first induced with arabinose and 4-HT or caffeine. They were then sorted for populations with fluorescence higher than the library without 4-HT or caffeine induction (positive sort). Sorted cells were regrown in the presence of arabinose only and then sorted for populations with lower fluorescence (negative sort). The positive sort was repeated once, and the retrieved cells were spread onto LB agar to obtain single colonies for picking.

**Fluorescence activated cell sorting (FACS) experiment**. For the initial sort, 100 μL of the library was inoculated into 25 mL of medium with inducers and grown overnight for 18–24 h. The next day, 10 μL of the culture was diluted into 10 mL 1× phosphate-buffered saline and then passed into the cytometer. Cell sorting was performed on a FACS Aria IIu cytometer (BD-Biosciences) with BD FACS Diva Software v6.1.3, through the red fluorescence channel (excitation 561 nm, emission 610/20 nm), under the Purity Mode. Cells were first gated on irregularly shaped FSC-A and SSC-A gates to exclude non-cellular materials, and then gated on boundaries defined by the previous libraries with or without induction. Gate sizes and positions were tailored to individual experiments. See Supplementary Fig. 28b for example gating strategies. Typically, 0.5–1 million gated events were collected into a 15 mL conical tube with 5 mL of LB supplemented with 1 % D-Glucose (10117, VWR) and without antibiotics. Collected cells were recovered for 2 h at 37 °C with 160 rpm shaking. Then, the volume was topped to 15 mL using LB with the next set of inducers and grown overnight for 16–18 h for the next sorting experiment. After the final cell sorting, the overnight culture was diluted and plated onto LB agar to obtain single colonies for strain isolation.

**Candidate strain isolation, characterization, and split/insertion site mapping**. In most cases >500 single colonies with desirable traits were individually picked into 96-well plates with 200 μL of LB medium for 16–18 h of growth, which were subjected to 16-24 h of induction assays to look for AND logic (mCherry, β-lactamase and ECF20), NAND logic (TetR and SrpR), or differential expression (with the M86 intein inserted or bisected). The fluorescence of the candidate clones was then measured on the FLUOstar Omega plate reader (BMG Labtech) and ranked. The best clones with desirable traits (~96 for IBM and ~40 for M86 intein engineering) were isolated with assistance from an OT-2 robot (Opentrons). The shortlisted clones were saved as temporary glycerol stocks and then subjected to fluorescence assays as described above for proper characterization, which generated the data for plotting the bisection/insertion maps. Strains that showed strong experiment-to-experiment variations in fluorescence were excluded for further use. A small aliquot of each cell strain in liquid suspension was then subjected to polymerase chain reactions (PCR), which amplified the N-terminal (mCherry) or the C-terminal (all others) joints. The PCR products were purified and sent for

Sanger sequencing. Poor sequencing results or reads that suggested non-single clones were discarded. The rest of the sequencing results were analyzed with a customized Python script utilizing the Biopython package v1.76[72] to deduce split or insertion sites by local alignment of sequences. Sites were mapped back to their fluorescence profiles and protein secondary structures (rendered using the Biotite package v0.20.1[73]).

**SDS-PAGE and Western blots for whole-cell lysate analysis**. Cells cultured for western blots were grown in 30 mL universal tubes (E1412-3011, Starlab) placed inside a shaker (Infors HT) maintained at 37 °C, 160 rpm. For each sample, a single colony was inoculated into 2 mL of medium and grown for 16–18 h. The next day, the overnight culture was diluted 1:100 into 2 mL of fresh medium with the appropriate inducers and grown for 24 h unless specified otherwise. Then, where appropriate, 0.5 μL of culture was removed for fluorescence measure for flow cytometry, with the pre-lysis fluorescent distributions displayed in the same figure. For detection of protein expression, in most cases 1 mL of culture was harvested. Exception to the above applies to the split mCherry-split M86 intein splicing experiment, the mCherry BiFC experiment and the split TetR-SYNZIP experiment, where the volumes of bacterial culture harvested were adjusted by optical densities to standardize the amount of cellular materials used in cell lysis. Bacterial cells were centrifugation at 17,000 × g and resuspended in 50 μL of 1× Laemmli sample buffer (1610747, Bio-Rad), boiled at 100 °C for 10 min, and centrifuged at 17,000 × g for 10 min. Five or ten microliter of the supernatant were resolved on an Any kD TGX Stain-Free protein gel (4568126, Bio-Rad) alongside a Chameleon Duo Pre-Stained Protein Ladder (928-60000, Li-cor). Protein contents were then transferred to a nitrocellulose membrane (1704270, Bio-Rad) through a semi-dry transfer protocol (1704150, Bio-Rad). The manufacturer's protocol (Doc. #988-13627, Licor) was followed for blocking, antibody incubation, washing, and detection of near-infrared probes on secondary antibody. We used 5% (w/v) skimmed milk in 1× tris-buffered saline (1706435, Bio-Rad) as the blocking reagent. The mCherry constructs involved in the western blot assays carried a hex-ahistidine tag at the C-termini and an epitope (residues 27–41) exists within the N-lobes. Bipartite proteins were detected using the rabbit anti-mCherry (A00682, Gen-Script, 1:3000 diluted), the mouse anti-His (A00186, GenScript, 1:5000 diluted), and the rabbit anti-HA (902303, Biolegend, 1:1000 diluted) antibodies. They reacted against the IRDye 680RD goat anti-rabbit (925-68071, Licor) or the IRDye 800CW goat anti-mouse (925-32210, Licor) secondary antibodies, both diluted at 1:20,000. Membrane imaging was performed on the Odyssey CLx Infrared Imaging System (Licor) and the resulting images were processed using Image Studio Lite v5.2.5 software (Licor) and ImageJ[74].

**Secondary structure alignments by amino acids and protein 3D structures**. Secondary structures of SrpR and ECF20 were predicted using the JPred4 Server[43], or modeled via SWISS-MODEL[75]. SrpR and TetR amino acid sequences were aligned with the known TetR structure (PDB: 4AC0) using PROMALS3D[44] with default parameters. 3D structures were rendered using the software PyMOL v1.7.6.7 (Schrodinger).

**Library preparation, next generation sequencing and data analysis**. Glycerol stocks of the final library were thawed and for each library, 200 μL of the stock was inoculated into 50 mL of fresh medium for overnight growth. Subsequently the plasmid DNA libraries were extracted by midiprep. For IBM libraries of mCherry, TetR, SrpR, and ECF20, appropriate combinations of restriction enzymes were used to release a minimal length of DNA fragments that contain mixed insertions at various positions. Digested DNA were resolved on agarose gels and the fragments with mixed insertions, which migrated as a single band, was excised and purified. For the domain-insertion library of ER-LBD into the M86 intein which was within mCherry, the region with insertions were amplified by PCR, then resolved and purified from agarose gels. Purified DNA were sent to Novogene (UK) for fragmentation and sequencing to obtain at least 7 million reads of 150 bp paired-end per library.

The resulting data was processed by in-house developed Python scripts. 12 bp, each at the 5′ and 3′ termini of the final inserted DNA were defined as signature sequences. Raw FASTQ files were filtered for reads that contained perfect matches to these signature sequences. Then, for each filtered read, the signature sequences were aligned and the adjacent sequence (12 bp) was extracted from the read, which were then aligned back to the CDS of the target protein to determine insertion positions. Only unique and perfect matches were considered authentic and a split/insertion site was called. Any fragments, where the forward and reverse reads reported different split/insertion sites were removed, and fragments where the forward and reverse reads pointed to the same split/insertion site were deduplicated to avoid double counting. Rare instances of sites mapped beyond the permitted transposition window were also removed. Similar to previous works, a productive split or insertion site on the amino acid sequence was called only if the insertion orientation was forward and the insert was in-frame.

**Data processing and statistics**. All data were processed and graphed in Python. Whenever displayed, fluorescence distributions shown within the same subpanel were normalized to their individual modes. We used two-tailed t-tests for independent samples assuming unequal variances in comparisons of fold changes

between median fluorescent values, and in comparisons of median fluorescent values from populations. Calculations were done in Python using the SciPy package v1.4.1[76]. Exact sample sizes ($n$) were described in figure legends, but in cases where sample sizes ($n \geq 3$) of different sites differed greatly between groups and were too numerous to report as exact values, we refer to Supplementary Data 2, sheet "sample_sizes" for exact values. Owing to the large number of statistical tests performed within a single figure panel, we did not report the individual statistics and $p$-values but rather the summary statistics: n.s. not significant; *$p \leq 0.05$; **$p \leq 0.01$; ***$p \leq 0.001$. The exact $p$-values can be found in Supplementary Data 2, under sheet "$p$-values".

**Reporting summary**. Further information on research design is available in the Nature Research Reporting Summary linked to this article.

## Data availability

Source data, including uncropped western blot images and Python scripts for generating figures, are deposited to the Edinburgh DataShare [https://doi.org/10.7488/ds/3001]. Uncropped western blot images are also present in a Source Data file. Raw sequencing data of IBM and DIM final libraries from NGS are deposited to the Sequence Read Archive under the project accession code PRJNA678813. List of constructs used in this study are detailed in Supplementary Data 1, and their sequences are available on SynBioHub[77] [https://synbiohub.org/public/Intein_assisted_Bisection_Mapping/Intein_assisted_Bisection_Mapping_collection/1]. Representative key constructs used in this study, which allow researchers to conduct IBM of their own, are deposited at Addgene (ID 161937–161955, see Supplementary Data 1 for details). Protein structures for analysis, including mCherry (2H5Q) [https://www.rcsb.org/structure/2H5Q], TEM-1 β-lactamase (1ZG4) [https://www.rcsb.org/structure/1ZG4], TetR (4AC0) [https://www.rcsb.org/structure/4AC0], and *Ssp* DnaB$^{M86}$ intein (6FRH) [https://www.rcsb.org/structure/6FRH] were assessed from the Protein Data Bank[rcsb.org][78,79]. Source data are provided with this paper.

## Code availability

Python scripts for analyzing Sanger sequencing results to determine split sites at the final step of IBM, and for analyzing split or insertion site coverages from NGS data, are available at GitHub at https://github.com/tyhho/IBM[80].

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

## Acknowledgements

We thank Dr Martin Waterfall from the Flow Cytometry Core Facility at the University of Edinburgh for performing cell sorting, Dr Hung-Ju Chang and Dr Jerome Bonnet for advice on acVHH-related experiments, Prof. Bryan C. Dickinson for providing plasmids encoding the evolved split T7 RNA polymerase, members of the Wang and Menolascina labs for helpful discussions, and Dr. Jamie Gilman and Dr. James Bryson for feedback on the manuscript. We would also like to thank the SBS BioRDM Team for their help with data deposits and data curation. This work was supported by the UK Research and Innovation Future Leaders Fellowship [MR/S018875/1], Leverhulme Trust grant [RPG-2020-241] and UK Biotechnology and Biological Sciences Research Council grant [BB/N007212/1] to B.W. T.Y.H.H. was supported by the Darwin Trust of Edinburgh and the Edinburgh Global Research Scholarship. A.S. was supported by a Microsoft Research PhD Scholarship.

## Author contributions

B.W. and T.Y.H.H. conceived the study. T.Y.H.H. developed the methods and designed the experiments with inputs from B.W. T.Y.H.H., A.S., and Z.L. performed the experiments. T.Y.H.H. analyzed the data with inputs from B.W., N.D., L.W. and F.M. H.S. provided the reagents for transposition. All authors took part in the interpretation of results and preparation of materials for the manuscript. T.Y.H.H. and B.W. wrote the manuscript with comments from all authors. B.W. supervised and acquired the funding of the study.

## Competing interests

The authors declare no competing interests.
