## [Peer Review File · Nature Communications]

Reviewers' Comments:

Reviewer #1:

Remarks to the Author:

My original opinion has not changed. The manuscript is certainly stronger but my opinion is that it is still an incremental advance in this arena. If someone wants to make a split-protein they already have many options and unclear if this adds a lot. Would not be my first choice as simply testing a couple of sites with appropriate CIDs is not difficult (clearly a preponderance of split-proteins that do nothing more than being split instead of being used). Also, if one wants complete coverage there are alternate methods.

Regardless, there are no major technical issues and overall well designed experiments. Should certainly be published somewhere (up to editors to decide) and time will tell if this is a method of choice for split-proteins - my simple opinion is that it is not.

Reviewer #2:

Remarks to the Author:

This revised manuscript is an improved version of the original with the new supplementary figures 7 and 14 in particular strengthening the paper claims. I also found that the response to the reviewers was valuable in explaining some of the points and encourage the authors to publish it alongside the manuscript when accepted.

Unfortunately I still think that the drug-induced splicing experiments are not compelling and may weaken the whole. I understand it is a proof of principle to show that the method can be extended to such applications but wonder if it is really necessary in this manuscript (although this would be the choice of the editor). Nevertheless, this manuscript describes a new general methodology to generate split-proteins that seems approachable enough to be taken up by the community, warranting publication.

Minor issues need to be fixed. Please check in particular references to supplementary figures in the main text as, for example, in the response to reviewer, the figure reporting the individual expressions of the N- and C-lobe of split Srp is labelled as Suppl. Fig 14 but is referred to as Suppl. Fig. 13 in the main manuscript. Note that this may not be the only case as for some reason I couldn't open the supplementary figures document on my computer.

Reviewer #3:

Remarks to the Author:

I thank the authors for their thorough replies to my many comments and requests.

I appreciate that they did new experiments, which have strengthened the paper.

I think the manuscript is now suitable for publication.

One last note: please, look at Fig. 10 in Palanisamy et al. You will see that the authors did identify five functional splice sites for puromycin acetyltransferase.

We thank the reviewers again for their comments, and again we provide our point-by-point responses. The original comments were colored grey and our responses were colored blue.

Reviewer #1 (Remarks to the Author):

My original opinion has not changed. The manuscript is certainly stronger but my opinion is that it is still an incremental advance in this arena. If someone wants to make a split-protein they already have many options and unclear if this adds a lot. Would not be my first choice as simply testing a couple of sites with appropriate CIDs is not difficult (clearly a preponderance of split-proteins that do nothing more than being split instead of being used). Also, if one wants complete coverage there are alternate methods.

Regardless, there are no major technical issues and overall well designed experiments. Should certainly be published somewhere (up to editors to decide) and time will tell if this is a method of choice for split-proteins - my simple opinion is that it is not.

We thank Reviewer #1 in recognizing our effort to improve our manuscript. We understand that a trial-and-error approach remains the popular approach with many biologists, and many would prefer to trust their intuitions or computational predictions. Still, we believe our method has decisive advantages, especially when 3D structures are not available, and that when globally optimal split sites are needed to maximize split protein performance. These are benefits not offered by existing methods, and hence we believe our method will become more prevalent in the future.

We would also like to apologize to Reviewer #1 for our mistake. Indeed, as pointed by Reviewer #3, there are 5 split sites for the gp41-1 intein in puromycin acetyltransferase in the paper by Palanisamy et al. We are sorry for being careless on this.

Reviewer #2 (Remarks to the Author):

This revised manuscript is an improved version of the original with the new supplementary figures 7 and 14 in particular strengthening the paper claims. I also found that the response to the reviewers was valuable in explaining some of the points and encourage the authors to publish it alongside the manuscript when accepted.

Unfortunately I still think that the drug-induced splicing experiments are not compelling and may weaken the whole. I understand it is a proof of principle to show that the method can be extended to such applications but wonder if it is really necessary in this manuscript (although this would be the choice of the editor). Nevertheless, this manuscript describes a new general methodology to generate split-proteins that seems approachable enough to be taken up by the community, warranting publication.

Minor issues needs to be fixed. Please check in particular references to supplementary figures in the main text as, for example, in the response to reviewer, the figure reporting the individual expressions of the N- and C-lobe of split Srp is labelled as Suppl. Fig 14 but is referred to as Suppl. Fig. 13 in the main manuscript. Note that this may not be the only case as for some reason I couldn't open the supplementary figures document on my computer.

We would like thank Reviewer #2 for the encouraging comments. In accordance with Reviewer #2's and the editor's suggestions, we have rewritten our manuscript, during which we de-emphasized and shortened the section on developing inducible inteins. We refocused on our major goal, which is to develop a method to identify insertions sites for split inteins. We hope this would address the concern raised.

We also apologize for the unprofessional mistake in referencing the wrong supplementary figure. This has been rectified in the rewritten manuscript, and this time we confirmed all supplementary figures are properly referenced to.

Reviewer #3 (Remarks to the Author):

I thank the authors for their thorough replies to my many comments and requests.

I appreciate that they did new experiments, which have strengthen the paper.

I think the manuscript is now suitable for publication.

One last note: please, look at Fig. 10 in Palanisamy et al. You will see that the authors did identify five functional splice sites for puromycin acetyltransferase.

We appreciate Reviewer #3 for the support on our revised manuscript, and we would like to thank Reviewer #3 for correcting us on the number of split sites on puromycin acetyltransferase.